# Exploring user acceptance of online virtual reality exhibition technologies: A case study of Liangzhu Museum

Jia Li[1], Chan Lv [2]*

1 Department of Environment Design, College of Design, Jiaxing University, Jiaxing, Zhejiang, China,
2 Department of Animation and Digital Media, College of Arts, Wuyi University, Wuyishan, Fujian, China

* 277540748@qq.com

**Data Availability Statement:** The data are available from Figshare (doi: https://doi.org/10.6084/m9.figshare.26181599.v3), and all relevant data are within the manuscript and its Supporting Information files.

## Abstract

Museums increasingly rely on cutting-edge digital technologies to attract visitors. Understanding the intricate factors influencing user acceptance of these technologies is, however, crucial for their effective use. This study therefore proposes a model, grounded in the technology acceptance model, to investigate user acceptance of online virtual reality (VR) museum exhibitions. Leveraging the online VR exhibition at Liangzhu Museum as a case study, data were collected from 313 participants and analyzed using partial least squares structural equation modeling (PLS-SEM) with Smart PLS. Semi-structured interviews with 15 individuals were conducted to complement the quantitative findings. The results reveal that factors such as interactivity, immersion, and presence positively influenced users' intrinsic technological beliefs (perceived ease of use, perceived enjoyment, and perceived usefulness), ultimately affecting their willingness to use and intention to visit on-site. Notably, immersion had a direct positive effect on perceived usefulness. There is a pressing need to leverage digital and web technologies to cater to the increasingly complex and diverse needs of online visitors, and emphasizing navigational performance in online VR exhibitions is also paramount for enhancing the overall user experience.

## Introduction

The development of the internet and digital technologies has revolutionized people's lifestyles, which has prompted museums to reassess their relationship with their audience [1]. With the widespread adoption of consumer-grade digital technologies and personal smart devices, individuals are increasingly reliant on mediums such as smartphones and computers to access information [2, 3]. Considering this trend, the traditional reliance of museums on physical visits is being called into question [4]. To tackle this challenge, many museums have proactively embraced the latest online digital technologies and collaborated with experts across various fields to develop diverse digital resources, including virtual reality (VR) [5]. These innovative digital technologies and virtual environments have emerged as crucial tools for enhancing museums' competitiveness and attracting new visitors [6]. By effectively harnessing online digital resources such as social media and virtual exhibitions, the dynamics of the relationship

**Funding:** This study was funded by Fujian Province Social Science Foundation Project approval number FJ2021C106. The funders had no role in study design, data collection and analysis, decision to publish, or preparation of the manuscript.

**Competing interests:** The authors have declared that no competing interests exist.

between museums and individuals, as well as between museums and society, are undergoing significant transformations [7]. However, the integration of digital technology into museums elicits a multitude of complex factors that can influence their success while posing challenges for assessing associated risks and leading to indecision, particularly among resource-limited museums [8].

The recent COVID-19 pandemic accelerated digital transformation in the museum sector while also exposing deficiencies in digital resources. The pandemic resulted in a reduction of physical visitors to museums and the closure of many museums, which prompted these institutions to expedite the development of online digital resources [9–11]. However, digital resources in museums still have limitations. One significant reason is the lack of close connection between museums' digital resources and their users [12]. The instability and complexity of digital technology can often lead many museums to outsource the development of digital resources to technical personnel and experts, which can result in a disconnect between the new technologies in the museums and the actual perceptions and needs of users [13]. Due to a lack of understanding among users of digital resources, many museums have found it challenging to determine which digitization methods are most effective [9]. Consequently, compared to the continuous development and integration of digital technologies in museums, in-depth user research and feedback mechanisms are still lacking, which may affect the effectiveness of the systems employed and the user experience [14, 15].

Within the context outlined above, the present study poses the following research question:

RQ: What are the primary factors affecting users' acceptance of online virtual exhibition technology in museums?

This study sought to achieve the following main objectives to promote the sustainable development of museum digital resources:

1. Based on the technology acceptance model (TAM), we established a structural equation model for visitors' acceptance of museum online virtual reality exhibitions and propose research hypotheses.

2. Taking the online VR exhibition adopted by the Liangzhu Museum as an example, we analyzed the factors influencing visitors' acceptance of this technology using structural equation modeling (SEM) and evaluated the relationships between them.

3. We then supplemented and extended the quantitative research by conducting interview studies.

The rest of this article consists of six major parts. The first part, the literature review, compiles the background information on current research in relevant fields through a review of literature related to the research topic. The second part presents the research model, the research hypotheses, and the main variables in the research model. The third part presents the methodology, primarily instrument development, research materials, data collection, and analysis tools. The fourth part covers the research findings, analyzes the collected data, and validates the effectiveness of the research model. The fifth part discusses quantitative and qualitative research results. The sixth part provides the research conclusion and study limitations.

## Literature review

### VR exhibitions in museums

VR, also known as a virtual environment, originated in the United States in the 1960s. In the decades since its inception, VR has played a significant role in enhancing educational and entertainment experiences [16]. Conceptually, VR traces its roots back to the popularity of

panoramas in Europe during the 19th century, according to Nedelcu [17]. Notably, Byerly [18] suggested that panoramas emerged from the Victorian fascination with tourism, offering an experience of "being in two places at once" while emphasizing a perceptual shift from knowingly experiencing an illusion to feeling it as reality. Thus, in some respects, VR exhibitions in museums can be considered the panoramas of the digital age [19].

The introduction of VR (in the modern sense) into museums began in the mid-1990s [20]. As the cost of 3D mapping software decreased, the barrier to creating virtual environments fell significantly, which facilitated the integration of VR technology into the cultural heritage sector [21]. Museum VR exhibitions have appeared in various forms, ranging from large-scale cave automatic virtual environments (CAVEs) to simple multimedia displays and applications [20]. The significance of VR exhibitions for museums lies in overcoming temporal and spatial constraints on visitors [20]. These constraints include limitations imposed by the museum's physical space, difficulties in reproducing vanished or inaccessible heritage, and restrictions on interacting with fragile and endangered exhibits[22, 23].

In recent years, VR based on panoramic photography has garnered attention from researchers and developers of museum VR exhibitions due to its simplicity and effectiveness [24]. Montagud and Orero [25] argued that, although panoramic-based VR cannot offer the same level of 3D interaction as fully virtual environments, it can directly capture real-world scenes to provide a strong sense of immersion.

## Museum VR exhibition user study

Early research has primarily focused on the general issues and methods of integrating VR into museum contexts, but systematic user studies are lacking. For instance, Lepouras and Charitos [26] identified visitor demands for VR exhibitions based on the quality of visitor experiences (e.g., immersion, display modes, resolution) and provided essential insights for related design and development. In a study of an immersive VR exhibition system for cultural heritage, Roussou [27] suggested that VR has the potential to offer museum visitors high-quality visual aesthetics and interactive experiences for information cognition, but the high application and maintenance costs cannot be ignored. Lepouras and Vassilakis [28] conducted a small-scale informal user evaluation study to test a prototype desktop-level VR exhibition system using low-cost 3D gaming technology. Carrozzino and Bergamasco [29] argued that VR stimulates visitors' senses through images, sounds, and other information in museums while allowing nonprofessional users to obtain information effectively from museum exhibitions.

With the continuous development of VR technology, its potential application in museums has been continuously explored, and user research has become more systematic in recent years. Izzo [30] highlighted the advantages of VR in terms of information richness and the customization of experience based on information and communication technology attributes. Su and Teng [31] conducted a user study on cross-object user interfaces in museum VR exhibition environments. One significance of this study is to demonstrate that user research findings for museum VR exhibitions may differ from the expectations of designers and developers, thus highlighting the necessity of user research. Robbani and Rosmansyah [32] argued that online VR based on panoramic photography, in conditions where physical visitation is not possible, can still enhance museum visitor experiences in terms of education, entertainment, immersion, overall experience, and visit intensity.

## Museum VR exhibition TAM study

Earlier studies have employed TAM to assess user acceptance of VR technology in museums [33]. Huang and Backman [34] further expanded the TAM by adding three exogenous latent

variables: computer self-efficacy, personal innovativeness, and media richness. They demonstrated the potential of digital virtual museum technology for enhancing experiences and learning. Although they repeatedly emphasized that VR technology is an integral part of virtual museums, they did not explicitly specify the type of technology they used for their case study.

Recent studies applying TAM to museum VR exhibition technology have further expanded the boundaries of research in this field and made it more adaptable to different museum research contexts. Hammady, Ma, and Strathearn [35] measured the acceptance of mixed reality (MR) technology for visualizing historical information in museums using TAM. However, the limited sample size and informal prototypes they used could restrict the validity of their findings. Wu et al. [36] used an extended TAM to investigate the factors influencing users' adoption of online digital virtual exhibitions in costume museums; however, their study sampled scholars and students from the fashion design field, who might not be representative of the general public's perspective. Iftikhar, Khan, and Pasanchay [37] assessed the acceptance of VR technology in tourism-related activities by people with disabilities based on the TAM theory. Wen, Sotiriadis, and Shen [38] identified key factors for visitors' acceptance and adoption of on-site VR technology in cultural heritage museums. However, their study, along with the aforementioned studies, lacks qualitative research to supplement the quantitative findings, which might lead to the omission of some key factors and details. As museums are a type of tourism resource, their research results have limited reference value. Therefore, given the increasing number of museums offering online VR exhibitions, research on user acceptance of this technology is necessary [39]. In summary, there is still a lack of mixed-methods research combining quantitative and qualitative approaches based on TAM theory, and specifically on the acceptance of VR exhibition technology in online museums.

## Research model

### TAM

TAM is a theoretical framework developed by Davis and Bagozzi [40] that has been widely employed in the study of user acceptance of new technologies. It is based on the theory of reasoned action (TRA), and the classic TAM comprises three main components: external variables, internal beliefs, and behavioral intention [41, 42] (Fig 1). Exogenous latent variables are associated with different users, technologies, and tasks. Technological beliefs include perceived ease of use and perceived usefulness, while user behavioral intention encompasses attitudes toward use, behavioral intention to use, and actual system use. Model analysis based on the TAM theoretical structure makes it possible to test the rationality of TAM variables and explore the causal relationships between variables, ultimately aiding in the explanation and prediction of user acceptance of new technological systems. TAM has more recently been widely applied in the study of various new technologies, including smart educational technology, the metaverse, and artificial intelligence, demonstrating its flexibility and effectiveness [43–48].

### External variables

Many studies have augmented external variables to adapt to different research contexts [49]. These variables are typically associated with specific tasks, types of technology, and user groups [50]. Drawing upon relevant theoretical literature, this study proposes three external variables: interactivity, immersion, and presence.

Interactivity is a technological characteristic of VR. It is defined as the ability for users to engage actively with the content of the VR system [51]. Human interaction with information technology includes both interactions with people and with information [52]. In the context of

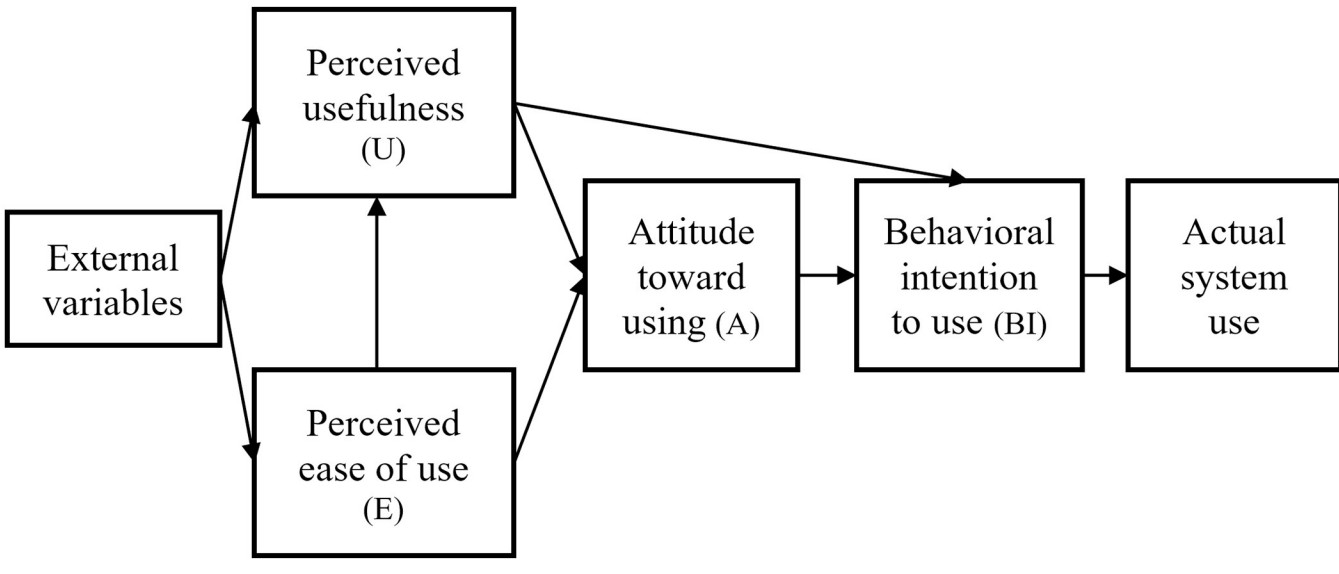

**Fig 1. TAM.**

this study, interactivity primarily refers to interaction with information, which can lead to positive responses toward information technology and its beliefs [52]. VR interactivity has been conceptualized as a characteristic that allows individuals to exercise control and engage in synchronous communication with the system [53]. A study on the impact of vividness and interactivity on VR experiences confirmed the importance of interactivity in VR experiences [51]. This study suggests that enhancing VR interactivity can enhance users' experience of media richness and further influence their future usage behavior. High levels of interactivity can, it has been suggested [54], enhance the sense of control over VR and lead to greater perceived enjoyment. The results of regression analyses indicate that high interactivity has a significantly positive impact on information acquisition and enjoyment of VR interfaces. Based on the above theories and experiences, this study considers interactivity as an important external variable for studying the acceptance of VR exhibition technology in museums. We therefore formulated the following hypotheses:

**H1**: Interactivity has a positive impact on perceived ease of use.

**H2**: Interactivity has a positive impact on perceived enjoyment.

**H3**: Interactivity has a positive impact on perceived usefulness.

Immersion can be understood as the user's lack of awareness of time and the real world while using a system, as well as the sense of involvement in and environmental feeling during tasks [55]. Jennett and Cox [55] emphasized the difference between immersion and presence, noting that an immersive experience is an experience of time, while presence is a state of mind. Bafadhal and Hendrawan [56] defined the VR immersive experience as a user's state of sensation and interaction in a virtual environment that is related to continuous sensory stimulation by VR system sensors; the higher the immersion, the more positive the user's attitude toward VR. Argyriou and Economou [57] suggested that immersion in VR based on panoramic photography involves the realism of the virtual environment, the degree of disconnection from the real world, and the perception of time. Furthermore, Vishwakarma and Mukherjee [58] argued that higher immersion can promote users' willingness to use VR systems and further enhance their perceived usefulness by increasing the enjoyment of use. Based on the above theories and experiences, this

study considers immersion as an important external variable affecting users' VR experiences, and formulated the following research hypotheses:

**H4**: Immersion has a positive impact on perceived ease of use.

**H5**: Immersion has a positive impact on perceived enjoyment.

**H6**: Immersion has a positive impact on perceived usefulness.

Presence can be understood as the feeling of "being there" in a virtual environment, and researchers have suggested that VR presence depends on the degree of isolation from real space in the virtual environment [59, 60]. Schubert [61] described presence in virtual environments as an unconscious feedback process of spatial perception, while Wu and Lai [62] also emphasized the spatial aspect of presence. This study posits that presence is a core concept based on panoramic photography VR systems that can replace or simulate a certain real-life environment; it is closely related to spatial intention. Research has indicated that a strong sense of presence can enhance users' emotional involvement in VR, predict their willingness to use the system, and reduce resistance to difficulties [63]. Tsai [60] pointed out that enhancing presence is an important method for improving visitors' cognition, emotions, and conceptual imagery related to the scene. In the context of on-site visits, studies in the field of online VR destination marketing have identified the influential role of presence in enhancing individuals' awareness of destination attributes [64, 65]. These studies affirm that VR presence can create positive user intentions toward destinations and promote their real-life visits. Based on the above theories and experiences, this study considers presence as an important external variable influencing visitors' acceptance of museum VR technology. The following research hypotheses were therefore formulated:

**H7**: Presence has a positive impact on perceived ease of use.

**H8**: Presence has a positive impact on perceived enjoyment.

**H9**: Presence has a positive impact on perceived usefulness.

## Internal belief variables

Perceived enjoyment is an internal belief variable developed by Davis and Bagozzi [66] based on the original TAM framework. Contrasting with perceived usefulness, which examines users' extrinsic motivation for using a new system, perceived enjoyment assesses users' internal motivation [66]. Extrinsic motivation refers to activities that contribute to increasing or adding additional value beyond the use of the system itself, such as acquiring information, enhancing skills, or increasing income. In contrast, perceived enjoyment refers to the degree of pleasure and joy experienced during the use of a new system, without any other foreseeable additional consequences. Davis and Bagozzi [66] argued that many new systems are rejected or considered unsatisfactory not because they fail to improve work performance or are difficult to use, but because they overlook the inherent enjoyment of system use. Explaining users' willingness and behavior to use new technology systems by perceived enjoyment is thus different from perceived ease of use and perceived usefulness.

Perceived enjoyment is one of the most widely applied variables in studies exploring the acceptance of new technology based on TAM [43, 48]. Some studies have examined the relationship between visitors' perceived enjoyment, perceived ease of use, perceived usefulness, and intention to use digital exhibition technologies in museums [35, 67]; other studies have further compared their differential impacts on intention to use [44, 48, 68]. This demonstrates the high potential of digital virtual technologies in creating a pleasant and enjoyable museum

experience [69]. Immersive and interactive digital virtual technologies generate more positive emotions and stimulate behavior [36]. However, the impact of perceived enjoyment has been controversial [33], so further exploration is needed.

Based on the above theories and experiences, the following hypotheses were developed:

**H10**: Perceived enjoyment has a positive impact on perceived ease of use.

**H11**: Perceived enjoyment has a positive impact on perceived usefulness.

**H12**: Perceived enjoyment has a positive impact on intention to use.

According to TAM, an individual's acceptance of new technology systems depends on the perceived usefulness and ease of use of those systems [70]. TAM assumes that users' acceptance of new technology systems is determined by their intention to use, which is influenced by their attitude toward use. This attitude is shaped by the basic technological beliefs of perceived ease of use and perceived usefulness [71, 72]. Perceived usefulness assesses the extent to which individuals subjectively believe that using a new technology system will benefit performance. Perceived ease of use assesses the effort individuals subjectively believe is required to achieve a goal using the new technology system [41]. According to Davis, perceived usefulness typically involves three items: job effectiveness, productivity and time savings, and importance. In contrast, perceived ease of use typically involves three items: physical effort, mental effort, and ease of learning. Many studies have, however, adjusted the scales for these concepts based on specific situations. For example, in a TAM study on cultural heritage VR conducted by Jung and Nguyen [73], perceived usefulness was emphasized in relation to the acquisition of information about cultural heritage sites. Based on the above theories and experiences, the following hypotheses were developed:

**H13**: Perceived ease of use has a positive impact on intention to use.

**H14**: Perceived ease of use has a positive impact on perceived usefulness.

**H15**: Perceived usefulness has a positive impact on intention to use.

## Behavior intention variables

This study draws on the research of Jung and Lee [74] and Hammady and Ma [35] to revise and integrate the impact of technological beliefs on users' attitudes, intentions, and behaviors into the intention to use, thereby reducing the behavioral intention construct in the research model. Promoting on-site visits is one of the main impacts of online VR technology [75]. Because online virtual experiences can only provide a small part of the on-site visit experience [76], virtual museums can be seen as a means to attract visitors to the physical museum site [77]. This point is of higher value in efforts to revitalize social vitality in the post-pandemic era [76]. This study therefore incorporated the tendency to visit actual sites into the behavioral intention construct and developed the following hypothesis:

**H16:** Intention of use has a positive impact on the tendency to visit actual sites.

The proposed model in this study, based on these hypotheses, is illustrated in the Fig 2.

## Methodology

### Instrument development

This study designed a research questionnaire based on the model described above and combined with results of previous research. The questionnaire consisted of three parts: a

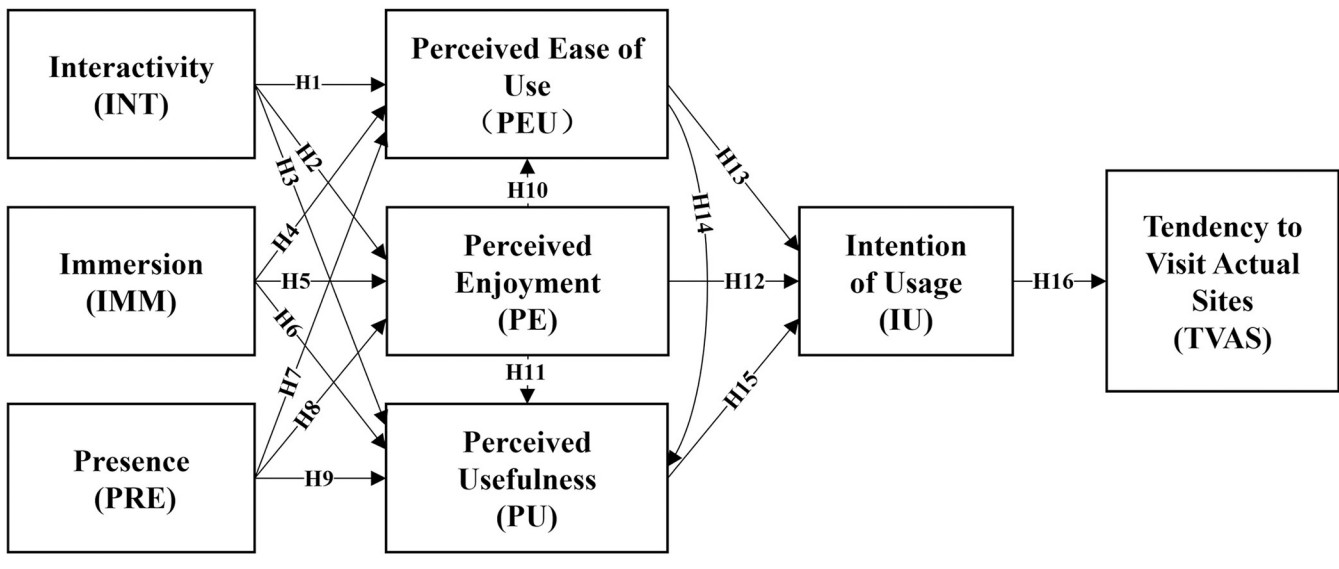

**Fig 2. Research model.**

demographic questionnaire, scale items, and an interview. Demographic variables included age, gender, and education level. The items in the scale questionnaire were all derived from published academic literature and modified according to the background of this study (Table 1). Items in the scale questionnaire were rated on a 5-point Likert scale, with response options ranging from *strongly disagree* to *strongly agree*. The interview portion was semi-structured, with questions designed based on the research model. Both the scale and interview questionnaire items were reviewed by three experts in the relevant field, and adjustments were made based on their feedback. The identities of the three experts are shown in the Table 2.

## Research materials

The online VR exhibition of the Liangzhu Museum was used as a case study, and its users were the research subjects. Located in Hangzhou, Zhejiang Province, China, Liangzhu Museum is a cultural heritage museum showcasing local Liangzhu cultural heritage in the form of archeological finds. Its main exhibits include various artifacts such as jade, pottery, stone tools, and lacquerware, as well as ancient human remains, plant and animal fossils, architectural and environmental models, and images of reconstructions.

The VR exhibition technology adopted by Liangzhu Museum uses an online VR exhibition system based on 360˚ panoramic photography. The permanent exhibitions of the museum were captured and produced through 360˚ photography, which allowed a simulated on-site visit experience in VR. Visitors can use their smartphones or computers to access the VR exhibition via the museum's official website (https://www.lzmuseum.cn/vr/index.html). This VR system possesses the following functionalities:

1. Simulated on-site visit: Users can switch between exhibition theme areas on their smartphone or computer screen to visit the exhibition. They can interact with the exhibit by, for example, zooming in or rotating the perspective. There are 63 theme areas available for exploration.

2. Audio guide: Automatic voiceovers play in certain exhibition theme areas, which provide an overall introduction to different exhibition themes. There are 19 audio guide points.

**Table 1. Questionnaire.**

| Construct | Scale Items | Source | Interview Items |
|---|---|---|---|
| Interactivity | INT1: The VR can respond quickly to my actions<br>INT2: I can feel myself interacting with the VR<br>INT3: The VR can understand and respond to my actions<br>INT4: I can control the VR perspective well<br>INT5: The VR is controllable | Lee and Lee [51]<br>Leung and Cheung [53]<br>Almogren, Al-Rahmi [78] | Q1: How do you feel about your interaction with this VR? |
| Immersion | IMM1: The VR keeps me focused on what I'm doing<br>IMM2: The VR immerses me in my tasks<br>IMM3: I am not easily distracted in the VR | Argyriou and Economou [57]<br>Jennett and Cox [55]<br>Vishwakarma and Mukherjee [58] | Q2: Do you feel immersed in this VR? |
| Presence | PRE1: It feels like I'm seeing the exhibition on-site<br>PRE2: It feels like being there, not just seeing it<br>PRE3: It feels like the return to reality is sudden when the VR ends | Bogicevic and Seo [64]<br>Huang and Backman [65] | Q3: Do you feel like you are entering another space in this VR? |
| Perceived usefulness | PU1: The VR allows me to quickly understand the features of Liangzhu Museum<br>PU2: The VR enhances the efficiency of my visit to the Liangzhu Museum<br>PU3: The VR provides useful exhibition information<br>PU4: The VR is important to me | Ahmad et al. [44]<br>Davis [41]<br>Hammady and Ma [35]<br>Jung and Nguyen [73] | Q4: Do you think this VR is useful? |
| Perceived ease of use | PEU1: Learning to use the VR is easy<br>PEU2: Becoming proficient in using the VR is easy<br>PEU3: I don't find experiencing VR exhibitions difficult | | Q5: Do you think this VR is easy to use? |
| Perceived enjoyment | PE1: I really enjoy experiencing the VR<br>PE2: I find the experience in the VR enjoyable<br>PE3: The experience in the VR is joyful | Huang and Backman [34] | Q6: In your opinion, does using this VR make you happy? |
| Intention of use | IU1: I might use similar VR exhibitions in the future<br>IU2: I plan to use similar VR in the future<br>IU3: Using similar VR in the future is important to me | Davis, Bagozzi and Warshaw [40]<br>Hammady, Ma, and Strathearn [35] | Q7: Will you use similar technology in the future? |
| Tendency to visit actual sites | TVAS1: After experiencing the VR, I want to know more about the actual site<br>TVAS2: After experiencing the VR, I am interested in visiting the actual site<br>TVAS3: After experiencing the VR, I plan to visit the actual site | El-Said and Aziz [76] | Q8: After using this VR, do you want to visit the museum in person? |
| Overall assessment | n/a | n/a | Q9: Overall, are you satisfied with this VR experience? |

Note: INT = interactivity, IMM = immersion, PRE = presence, PU = perceived usefulness, PE = perceived enjoyment, PEU = perceived ease of use, IU = intention of use, TVAS = tendency to visit actual sites.

**Table 2. Experts participating in questionnaire review.**

| | Specialty | Position |
|---|---|---|
| Expert 1 | Expertise in the field of commercial design, statistics, and over 9 years of quantitative research in the field of art design | PhD in Department of Business Administration, Chang Gung University |
| Expert 2 | Expert in virtual exhibition technology, with over 10 years of practical experience and research in virtual exhibitions for museums and exhibition halls | Head of Technology and Product Development at a VR technology company |
| Expert 3 | Expert in applied mathematics and statistics, with over 8 years of research and teaching in statistics | PhD in Applied Mathematics and Statistics from Huazhong University of Science and Technology<br>Full-time lecturer in the Mathematics Department of a university |

3. Video playback: Videos are played on virtual screens within the VR environment. These provide further information about specific theme areas. There is one video playback point.

The visual content of the VR exhibition consists of the permanent exhibitions of Liangzhu Museum, with different exhibition halls and theme areas represented by thumbnail icons at the bottom of the interface. Visitors can follow the on-site visit sequence by clicking on the guiding arrows in the picture, or they can switch to different theme areas by clicking on the thumbnail icons at the bottom. Background music plays throughout the visit.

## Data collection

Data were obtained from participants through scale-based questionnaires and semi-structured interviews. The target population of this study consists of ordinary users in China. All participants were recruited through social media platforms, ensuring that participation in the study was open to anyone. The recruitment period began on January 4, 2023, and ended on December 10, 2023. A mobile phone–based online questionnaire was employed to apply the scale items. Prior to completing the questionnaire, participants were required to use their mobile phones to explore the VR exhibition of the Liangzhu Museum and locate a specific designated exhibit, as shown in Fig 3. Once found, participants were free to explore the VR exhibition until they decided to end the session. We randomly selected a certain number of participants and asked if they would be willing to participate in further interviews.

Random sampling was utilized to contact a certain number of participants through social media, asking them if they were willing to participate in further interviews. If the participants agreed, they proceeded to the interview phase. Participants who completed all scale items were eligible to participate in a lottery draw. Participants who completed the interviews received an additional gift valued at approximately US$7.

Over a period of approximately 12 months, a total of 347 questionnaires were collected, of which 314 were deemed valid. The sample size thus met the minimum requirements for SEM, with each estimated parameter having between 10 and 20 cases [79]. A total of 15 participants were involved in interviews (9 females and 6 males). All participants and interviewees participated anonymously after providing informed consent, and recordings were made with the interviewees' consent.

## Analysis tools

This study used the partial least squares (PLS) method to analyze the data, with Smart PLS 2.0 as the analysis software. PLS is a computational method for structural equation modeling (SEM)—that is, PLS-SEM. According to relevant statistical data, there has been a trend of significant growth in the use of PLS in empirical research, which indicates that it is increasingly a mainstream method for SEM analysis [80]. The advantages of its low requirements for residual distribution and smaller sample size have contributed to its increasing application in recent years [81]. The sample size requirements for PLS analysis are generally based on the 10 times rule, which suggests that the minimum sample size should be 10 times the total number of paths involving exogenous and endogenous variables in the model [82–84]. In this study, the model comprised 16 paths, so the minimum sample size required was no less than 160.

## Ethical statement

The study was approved by the Academic Ethics Committee of Jiaxing University. The study complied with IRB principles. Prior to formally completing the questionnaire, all participants, including the guardians of minors, were duly informed regarding the purpose, content, data

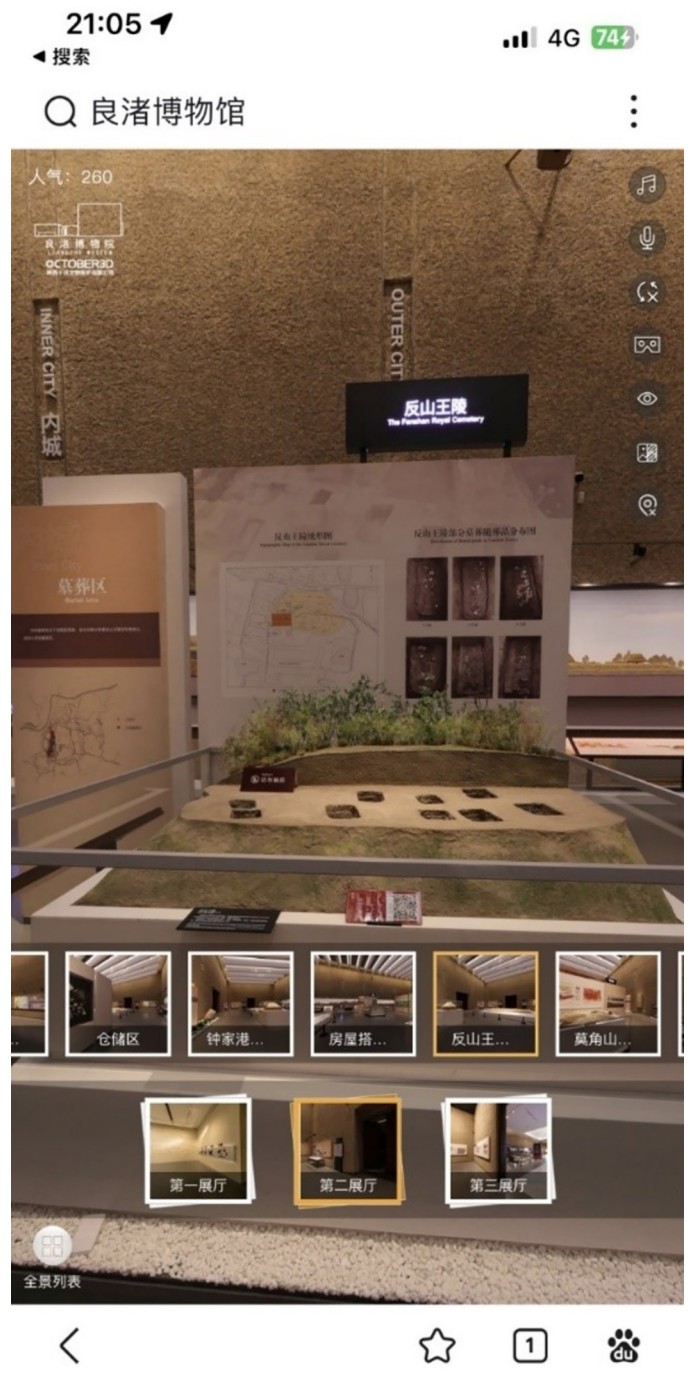

**Fig 3. The target exhibit that participants had to find.**

usage, risks, and benefits of the study, and all participants provided their electronic consent. All tests were done with the participants' consent, and all questionnaires and interviews were completed anonymously.

# Findings

## Sample description

This study collected a total of 313 samples, including 182 females and 131 males, resulting in a female-to-male ratio of 1.38:1. The age group of 30–39 had the highest representation, with 170 individuals, accounting for 54.31% of the total sample. The next most represented age group was 18–29, with 107 individuals, comprising 34.19% of the total sample. The remaining age groups were as follows: 40–49 (25 individuals, 7.99% of the total sample), 50–59 (10 individuals, 3.19% of the total sample), and under 18 (1 individual, 0.32% of the total sample). The majority of participants held a university degree (247 individuals, 78.91% of the total sample), followed by those with other professional certificates (30 individuals, 9.58% of the total sample) and those with postgraduate degrees (26 individuals, 8.31% of the total sample). The lowest educational attainment was high school or below, with only 10 individuals, comprising 3.19% of the total sample.

## Measurement model

We used factor analysis to evaluate the convergent and discriminant validity of the model. Convergent validity refers to the degree of association among observed variables within each construct in a research model to assess the rationality of item design within constructs [83]. Using the PLS algorithm, Cronbach's α coefficient, composite reliability (CR), and average variance extracted (AVE) were computed to test the convergent validity of each construct, thus measuring the internal consistency within constructs. Constructs were considered to have high internal consistency when the Cronbach's α coefficient was greater than 0.7, CR was greater than 0.7, and AVE was greater than 0.5. Table 3 presents the results of the convergent validity calculations for each construct. Most constructs had Cronbach's α, CR, and AVE values exceeding the acceptable thresholds (Cronbach's α, CR > 0.7, AVE > 0.5). Although the Cronbach's α for interactivity is 0.687, which falls slightly below the 0.7 threshold, it still meets the acceptable range of 0.6–0.7 and is close to 0.7 [85]. The results thus indicate good internal consistency within the constructs.

To further test the validity of the model, we employed cross-loadings and the Fornell–Larcker criterion to examine discriminant validity. It is generally considered that within the cross-leading of each construct, if the loading of each observed variable is greater than that of the other observed variables and exceeds 0.7, then the construct can be considered to have good discriminant validity. The Fornell–Larcker criterion assesses the discriminant validity between latent constructs by comparing the square root of the AVE of a construct with its correlations with other constructs [86]. The results for the cross-loadings and Fornell–Larcker criterion are presented in the Tables 4 and 5. The results indicate that the constructs demonstrate good discriminant validity.

**Table 3. Convergent validity results.**

| Constructs | Cronbach's α | CR | AVE |
|---|---|---|---|
| Perceived enjoyment | .752 | .858 | .668 |
| Immersion | .828 | .879 | .593 |
| Interactivity | .687 | .827 | .615 |
| Perceived ease of use | .719 | .842 | .640 |
| Presence | .798 | .881 | .712 |
| Perceived usefulness | .742 | .853 | .659 |
| Intention of use | .783 | .860 | .606 |
| Tendency to visit actual sites | .711 | .839 | .634 |

**Table 4. Cross-loading results.**

|  | IMM | INT | IU | PE | PEU | PRE | PU | TVAS |
|---|---|---|---|---|---|---|---|---|
| IMM1 | **.834** | .626 | .404 | .420 | .335 | .504 | .553 | .428 |
| IMM2 | **.827** | .620 | .389 | .402 | .282 | .593 | .550 | .393 |
| IMM3 | **.790** | .588 | .357 | .274 | .213 | .484 | .532 | .338 |
| INT1 | .612 | **.834** | .450 | .409 | .419 | .551 | .628 | .503 |
| INT2 | .618 | **.788** | .426 | .452 | .286 | .547 | .550 | .378 |
| INT3 | .545 | **.739** | .334 | .316 | .375 | .486 | .501 | .367 |
| INT4 | .567 | **.705** | .410 | .404 | .284 | .454 | .495 | .412 |
| INT5 | .539 | **.780** | .461 | .321 | .408 | .486 | .519 | .397 |
| IU1 | .420 | .484 | **.823** | .584 | .447 | .523 | .542 | .621 |
| IU2 | .370 | .403 | **.815** | .541 | .460 | .413 | .495 | .540 |
| IU3 | .307 | .380 | **.709** | .498 | .336 | .329 | .368 | .447 |
| PE1 | .370 | .399 | .569 | **.822** | .323 | .450 | .473 | .596 |
| PE2 | .392 | .437 | .559 | **.778** | .320 | .403 | .481 | .503 |
| PE3 | .325 | .352 | .532 | **.799** | .384 | .497 | .471 | .551 |
| PEU1 | .286 | .372 | .489 | .391 | **.844** | .402 | .425 | .470 |
| PEU2 | .276 | .379 | .402 | .350 | **.850** | .365 | .421 | .381 |
| PEU3 | .306 | .416 | .453 | .341 | **.837** | .351 | .449 | .469 |
| PRE1 | .527 | .551 | .448 | .463 | .377 | **.827** | .631 | .451 |
| PRE2 | .496 | .508 | .415 | .431 | .403 | **.818** | .597 | .479 |
| PRE3 | .554 | .544 | .470 | .479 | .292 | **.791** | .531 | .490 |
| PU1 | .551 | .589 | .501 | .491 | .374 | .589 | **.794** | .493 |
| PU2 | .503 | .543 | .461 | .475 | .394 | .555 | **.791** | .454 |
| PU3 | .496 | .524 | .460 | .470 | .392 | .557 | **.759** | .535 |
| PU4 | .525 | .528 | .459 | .411 | .436 | .551 | **.769** | .444 |
| TVAS1 | .375 | .406 | .522 | .553 | .407 | .400 | .491 | **.797** |
| TVAS2 | .395 | .455 | .560 | .517 | .467 | .472 | .503 | **.779** |
| TVAS3 | .366 | .421 | .563 | .573 | .377 | .513 | .484 | **.812** |

Note: INT = interactivity, IMM = immersion, PRE = presence, PU = perceived usefulness, PE = perceived enjoyment, PEU = perceived ease of use, IU = intention of use, TVAS = tendency to visit actual sites.

**Table 5. Fornell–Larcker results.**

|  | IMM | INT | IU | PE | PEU | PRE | PU | TVAS |
|---|---|---|---|---|---|---|---|---|
| IMM | **0.817** |  |  |  |  |  |  |  |
| INT | 0.749 | **0.770** |  |  |  |  |  |  |
| IU | 0.470 | 0.542 | **0.784** |  |  |  |  |  |
| PE | 0.453 | 0.495 | 0.692 | **0.800** |  |  |  |  |
| PEU | 0.343 | 0.461 | 0.533 | 0.428 | **0.844** |  |  |  |
| PRE | 0.646 | 0.657 | 0.546 | 0.563 | 0.443 | **0.812** |  |  |
| PU | 0.667 | 0.703 | 0.605 | 0.594 | 0.512 | 0.724 | **0.778** |  |
| TVAS | 0.475 | 0.537 | 0.690 | 0.688 | 0.524 | 0.581 | 0.619 | **0.796** |

Note: INT = interactivity, IMM = immersion, PRE = presence, PU = perceived usefulness, PE = perceived enjoyment, PEU = perceived ease of use, IU = intention of use, TVAS = tendency to visit actual sites.

**Table 6. Results of structural model.**

| Hypothesis | Path coefficient | t-value | p-value | Result |
|---|---|---|---|---|
| H1 (INT → PE) | .195 | 2.504 | .013* | Supported |
| H2 (INT → PEU) | .315 | 3.782 | .000** | Supported |
| H3 (INT → PU) | .215 | 2.697 | .007** | Supported |
| **H4 (IMM → PEU)** | **-.110** | **1.279** | **.202** | **Not supported** |
| **H5 (IMM → PE)** | **.045** | **.539** | **.590** | **Not supported** |
| H6 (IMM → PU) | .182 | 2.492 | .013* | Supported |
| H7 (PRE → PEU) | .184 | 2.293 | .022* | Supported |
| H8 (PRE → PE) | .405 | 5.214 | .000** | Supported |
| H9 (PRE → PU) | .306 | 3.871 | .000** | Supported |
| H10 (PE → PEU) | .219 | 3.105 | .002** | Supported |
| H11 (PE → PU) | .173 | 2.715 | .007** | Supported |
| H12 (PE → IU) | .471 | 8.559 | .000** | Supported |
| H13 (PEU → IU) | .223 | 4.187 | .000** | Supported |
| H14 (PEU → PU) | .141 | 2.982 | .003** | Supported |
| H15 (PU → IU) | .211 | 3.479 | .001** | Supported |
| H16 (IU → TVAS) | .690 | 16.311 | .000** | Supported |

*P≤.05 **P≤.01

Note: INT = interactivity, IMM = immersion, PRE = presence, PU = perceived usefulness, PE = perceived enjoyment, PEU = perceived ease of use, IU = intention of use, TVAS = tendency to visit actual sites.

## Structural model

This study employed bootstrapping to conduct structural model analysis of the research model to examine the causal relationships between latent constructs [83]. Through bootstrapping, the t-values and p-values of the paths were obtained to test the significance of path coefficients between latent constructs and validate the hypotheses. The computed results and decisions are summarized in Table 6. The new relationships among variables in the research model are depicted in Fig 4.

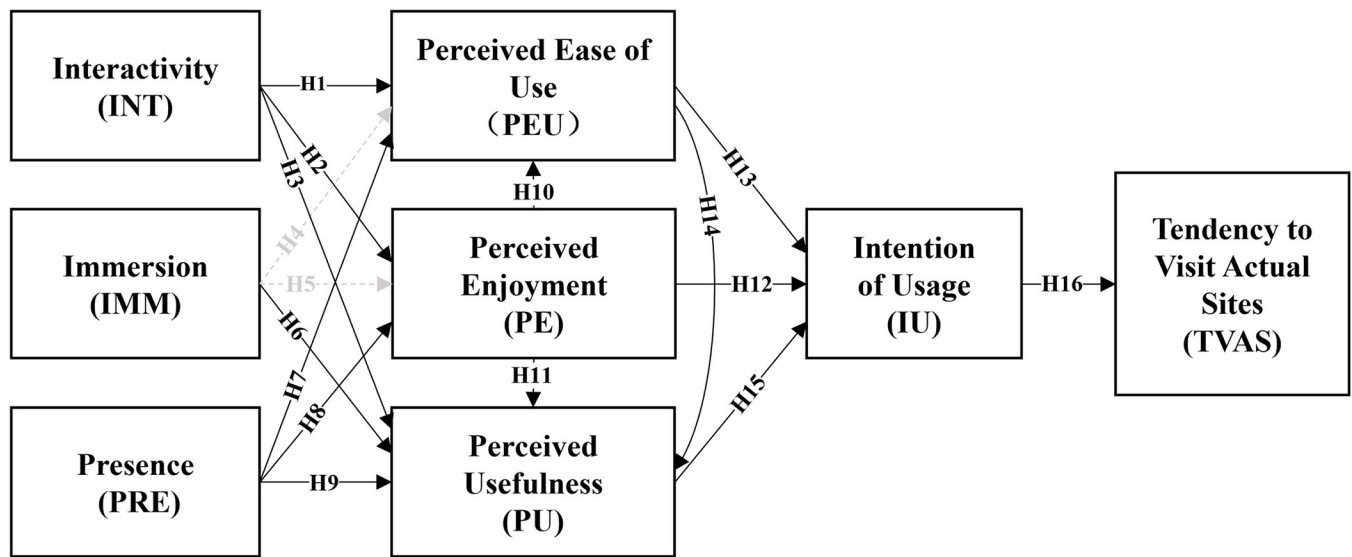

**Fig 4. Research model computed through structural modeling; gray paths indicate unsupported hypotheses.**

## Interview results

**External variables.**   For the interactivity construct, participants generally provided positive evaluations of the VR exhibition overall, but also raised some specific issues. Most participants indicated that it was "okay" (VF1, VM7, VM9), "not bad" (VF2, VF11), or "alright" (VM10). The issues raised by participants mainly focused on the perception of orientation and the diversity of interactions within the VR environment. For example, some participants (VF1, VM4, VM9, VF15) found it easy to become disoriented or lost during the visit, while others (VM4, VF5) felt that the interaction with exhibits and artifacts was too simple and wanted more interactive elements.

Although most evaluations of immersion were positive, many participants also pointed out existing issues. Some participants mentioned that immersion was affected by the small size of the phone screen or unclear content, which resulted in evaluations such as "average" (VF1), "not good enough" (VF9), and "occasionally there" (VF12). Some participants also provided an "average" evaluation (VM4, M9) due to issues with orientation perception. It is worth noting that some participants believed that the VR exhibition's music, voiceovers, and visual effects were good and enhanced the sense of immersion (VF5, VF11, VF14).

Most participants gave positive evaluations to presence. Many participants expressed feelings of "being there" (VF1, VF2, VF3, M9, VF12, VM14) or that the sense of presence was "very obvious" (VF6) and "like being there" (VF8, VF2). A significant number of participants attributed the good sense of presence to the realism of the VR visuals (VF1, VF3, VF12, VF13, VF2). However, some participants mentioned differences between the VR experience and the real world due to screen distortion (VM4) or lack of realism (VM7). Some participants also felt that presence was reduced due to a lack of interaction and diverse exhibition formats (VF5, VF15).

**Internal belief variables.**   All participants provided positive evaluations for the perceived usefulness construct. Almost all participants believed that the VR exhibition was useful and gave responses explicitly indicating its usefulness. However, some participants pointed out that the images and text were not clear enough and required extra effort to view (VF1, VF12). Others felt that the information provided by the VR was not only insufficient but also lacked firsthand data (VF6).

Most participants perceived the VR exhibition to be simple in terms of perceived ease of use. However, it is worth noting that some participants believed that it took some time to adapt before finding it simple (e.g., VF2, VM4, VM3.123, VF11). Some participants also mentioned that the small text of the user interface or the lack of prompts could affect perceived ease of use (VF3). Issues with orientation perception could also cause usability issues (VF8, VM9).

Most participants gave positive evaluations to the perceived enjoyment construct. Some participants mentioned that learning from the VR exhibition brought them joy (VF11). Interestingly, some participants thought that the VR music effects were excellent and enhanced the perceived enjoyment (VF2, VF3, VF13, VM8); however, there were also participants who felt the opposite (VF9). VF3 and M10 suggested that implementing gaming features in the VR arcade of the museum could further enhance enjoyment. It is also worth noting that perceived enjoyment of VR may decrease over time with prolonged experience (VF4, VM8, VF15).

**Behavior intention variables.**   In the intention of use construct, participants expressed varying degrees of willingness to use similar technologies in the future. When asked about their future use of similar technology, some participants gave very clear positive responses. However, there were also participants who believed that it would depend on the situation (VM5, VF11, VF15)—for example, they would use it before doing homework or traveling (VF12).

For the tendency to visit actual sites construct, although most participants admitted to having the intention to visit physically, the reasons varied. Some participants thought it was due to the lack of clarity in the VR content (VF1) or insufficient information that led to the desire for a physical visit (VM4, VM6, VF12). Others became interested in a physical visit due to their positive experience with the VR exhibition (VF3, VM7, VF13, VM14). Some participants were originally interested in museum visits and this led to their intention to pursue a physical visit (VF5). Additionally, some participants considered physical tourism due to the lifting of COVID-19 restrictions (VF5, VF11).

**Overall assessment.** The majority of participants provided positive evaluations regarding their overall satisfaction with VR. Many participants gave moderately positive overall ratings. However, some participants considered the current exhibition format to be relatively monotonous and thus provided average evaluations (VF15).

## Discussion

### Extension of existing research

This study found that the interactivity experience of museums' online VR exhibition positively influenced visitors' perceived enjoyment, perceived usefulness, and perceived ease of use. The results of this study confirm the significant role of interactivity in enhancing visitor acceptance of museums' online VR exhibition technology. The findings supported hypotheses H1, H2, and H3. This research thus corroborates and extends existing research findings, such as those from studies suggesting that interactivity can enhance users' perception of the richness of VR media and emphasize users' ability to control their experience [53, 87].

The immersion of the VR exhibition was positively correlated with perceived usefulness but did not positively influence the perceived ease of use and perceived enjoyment of the system. The immersive quality of VR can be understood as the extent to which the virtual environment provides users with a sensory experience close to objective reality that allows users to temporarily forget about the real world and the passage of time during the experience. According to the results of this study, hypothesis H6 was confirmed. This is consistent with some prior research conclusions; for example, Vallade and Kaufmann [88] suggested that if participants perceive the VR environment as real, they should consider it a high-quality experience, which affects users' perception of system usefulness.

On the other hand, hypotheses H4 and H5 were not supported by the results. This indicates that enhancing the immersion of museum online VR systems did not positively influence visitors' perceived ease of use and perceived enjoyment. This result is inconsistent with some existing research findings. For example, Vishwakarma and Mukherjee [58] suggested that a high level of immersion enables people to enjoy the online VR experience and increases their willingness to further use such technology. The analysis in this study suggested that the following reasons may lead to the lack of support for hypothesis H5:

1. The immersion of VR is determined by the extent to which VR systems stimulate users' senses to approximate the real objective environment. The difficulty of VR use is related to the complexity of its logic. A sense of immersion is obtained from the sensory experience of directly stimulating the senses, which is quite different from the rational experience of the VR usage logic, so there may be no direct relationship between the two.

2. According to the interview results, the perception of enjoyment in VR use is subjective and unstable. This may affect the relationship between immersion and enjoyment. For example, participant VF15 suggested that the perception of enjoyment in experiencing VR diminishes as the experience time increases. Some participants (VF5, VF11, VF14) thought that

music was one of the main things affecting immersion; however, according to the interviews, participants' evaluations of the music elements in VR were inconsistent. For example, participant VM9 explicitly stated a dislike for the existing VR background music. These reasons may lead to the lack of support for hypothesis H5.

The results of this study indicate that the sense of presence in the online VR experience positively influenced visitors' perceived ease of use, perceived enjoyment, and perceived usefulness. The sense of presence involves users' subjective psychological state of perceiving themselves as being present in a virtual environment, which can be briefly described as the degree to which users feel immersed in VR. This can be assessed by the extent of visitors' transition between the real world and the digital virtual world [49–51]. The results of this study supported hypotheses H7, H8, and H9. These conclusions also corroborate existing research findings [88–90].

The results of this study suggest that perceived enjoyment positively influences perceived ease of use, perceived usefulness, and intention of use in VR exhibition. Perceived enjoyment refers to the degree to which individuals feel happy when experiencing museum online VR exhibition technology, and this perception is unrelated to utility [91]. The results of this study supported hypotheses H10, H11, and H12. These conclusions are similar to existing research findings [92, 93]; however, in contrast to most studies that suggest a positive influence of perceived ease of use on perceived enjoyment, this study found that perceived enjoyment also has a positive relationship with perceived ease of use. For example, some research suggests that perceived ease of use is an important predictor of people's acceptance of new digital technology systems and directly affects their perception of system enjoyment [76]. This point has also been confirmed by many other studies [34, 94].

## New findings of this study

The design of online VR exhibition content in museums needs to consider the needs of various media and diverse audience groups. Although VR provides panoramic guidance composed of thumbnail images of different venues and exhibition areas, some respondents raised concerns about the small text size and incomplete display, which may affect their overall experience (VF3, VM14). One possible reason for this situation is that the VR test in this study primarily used smartphones as the main playback medium, which resulted in limited screen size. Comparing the playback effects between smartphones and computers, that VR text displayed on computers was indeed clearer and more complete. However, targeting smartphones as a platform has become a trend as smartphones have become the mainstream way for the general public to access information. Designers and managers of online VR museums should thus fully consider the compatibility between online VR exhibitions and smartphone display characteristics; they should develop graphic and textual content suitable for playback on smartphones.

Another aspect is the need to consider the differences in needs among elderly people, those with poor vision, and visitors who do not speak Chinese in the context of aging societies and globalization. Although the current experiment did not involve elderly people or those with poor vision, a considerable number of respondents (VF1, VF3, VM14) still mentioned that reading information was difficult due to small font sizes or graphics. The universality of VR is thus an important issue that cannot be ignored in the design, development, and deployment of VR applications.

The potential of digital and network technologies needs to be further leveraged to enrich the visitor experience in museum VR exhibitions. Online VR exhibitions are novel digital technologies that integrate video, music, and narration to provide visitors with a richer audiovisual

experience. This advantage was confirmed in interviews with respondents VF3, VM4, VM10, VF11, and VF13. However, many respondents also indicated that the online VR exhibition of the Liangzhu Museum did not fully meet their needs. For example, there was a lack of variety in the interaction between visitors and museum artifacts. Some respondents pointed out that, while VR offers gaming spaces, the current exhibition lacked gaming features (VF3, VM10), and there was limited interaction with exhibits (M7, F12). Additionally, it is worth noting that there were differing opinions about the background music for the museum's online VR exhibition (VF2, VF13, VF9), which indicates the complexity of visitor demands. To meet the needs of different visitors, it is necessary to increase the diversity and selectivity of exhibition content. Managers and developers should fully utilize the advantages of digital and network technologies to provide diversified services. These measures may include but are not limited to:

1. Providing hyperlinks to expand on artifacts, displaying more detailed and professional academic information to meet the needs of visitors with academic interests;

2. Providing more interactive and game designs to meet entertainment needs;

3. Providing more video, image, and text-based information to meet learning needs; and

4. Integrating the exhibition with social software or media to meet visitors' social needs.

Good navigational performance ensures that visitors can always understand their location and orientation, which is an factor for improving the acceptance of museum VR exhibitions. According to descriptions from respondents VF1, VM4, VM9, and VF15, it was easy to become disoriented and even experience circling in the virtual environment, which greatly affected their visitor experience. The analysis suggested that one reason for this could be the inconsistency between the change in screen perspective and the expected direction indicated by the VR guidance arrow after clicking on the arrow. Another reason could be the lack of a navigation system design adapted to smartphone screens; the existing navigation system icons and text were too small and incomplete on the phone, which made them easy to ignore. Future VR design and development or improvement should strengthen the design of spatial orientation and guidance systems.

## Research implications
### Theoretical implications

This study expanded the applicability and explanatory power of the traditional TAM by incorporating variables such as interactivity, immersion, and presence. Through a combination of quantitative and qualitative analyses, this study revealed how the experience of users of online VR museum exhibitions influences their intrinsic technological beliefs, thereby affecting their intention to use the technology and visit the museum. This extension not only enhances our understanding of user acceptance of online VR museum exhibition technology, but also fills a research gap in this field. This study also emphasized the need to consider different media and diverse user groups in VR content design, which can provide new insights for future research on the role of different device platforms in VR experiences. These findings offer new theoretical perspectives and empirical support for the further exploration of user experience with museum digital resources in academia.

### Practical implications

The research findings provide guidance for museums in formulating digital strategies and enhancing user experiences. First, museums should focus on improving the interactivity,

immersion, and presence of online VR exhibitions to increase user acceptance of the technology and their willingness to use it. Second, curators, exhibition designers, and other decision-makers should enhance the inclusivity and diversity of VR exhibitions to meet the needs of elderly individuals, those with visual impairments, and visitors who speak different languages. This can be achieved by offering interactive game designs and rich multimedia content to improve user experiences. Additionally, emphasizing the importance of navigability in online VR exhibitions can help enhance the overall user experience. This has practical significance for designing and developing online VR exhibitions based on 360˚ panoramic photography, especially in large and complex museum spaces. Through these improvements, museums can not only expand their audience reach and promote cultural dissemination, but they can also enhance the educational effectiveness of online exhibitions, enabling visitors to understand and experience cultural heritage more profoundly in a virtual environment.

## Conclusions and limitations

This study took the VR exhibition technology of Liangzhu Museum as an example and established a TAM-based research model after a literature review. The classic TAM was adjusted by adding exogenous latent variables, internal belief variables, and behavioral intention variables. The exogenous latent variables were interactivity, immersion, and presence. The internal belief variables were perceived usefulness, perceived enjoyment, and perceived ease of use, while the behavioral intention variables were intention of use and tendency to visit actual sites. The online VR exhibition of Liangzhu Museum was then used as a case study to verify the hypotheses about the variables and their relationships proposed in the model. Except for the lack of a positive impact of interactivity on perceived ease of use and perceived enjoyment, all other research hypotheses were confirmed. The research model demonstrated good internal consistency and predictive power. Additionally, navigability may be a new variable influencing visitors' acceptance of museum VR exhibitions.

Although museum online VR exhibitions have the advantage of digital technology, they still need to fully consider the differentiated needs of users. The interviews revealed that there are significant differences in visitors' demands for VR exhibitions in museums. While digital and network technologies expand the boundaries of museum visitors, they also increase the difficulty of understanding them. As Lester [95] pointed out, virtual exhibition visitors are far more diverse than traditional museum visitors; these differences may stem from their abilities, perceptions, and demands related to museums and digital virtual exhibitions, as well as the complexity of deeper cultural and social backgrounds. Understanding these complexities has long-term significance for promoting the sustainable development of museums, especially in the context of social diversification, democratization, and economic recovery.

Subsequent research should provide more case studies of different types of museums. Liangzhu Museum is a medium to large cultural heritage museum that primarily display local archaeological artifacts; it represents just one of the many types of museums in the world. Whether the conclusions of this study are applicable to other types and sizes of museums requires more interactive confirmation through different case results by subsequent researchers. More studies on different types of visitors are also needed. As mentioned, virtual exhibition visitors have diverse differences. Subsequent research should consider samples of different types of visitors, including families, tour groups, the elderly, children, and people with different physical and mental abilities. Subsequent research needs to understand these different visitor types more meticulously and deeply, particularly when confronted by VR exhibitions, to further enhance visitors' acceptance of this technology.

## Supporting information

**S1 Data. Data for statistical analysis.**
(XLSX)

## Author Contributions

**Conceptualization:** Jia Li.

**Data curation:** Jia Li, Chan Lv.

**Formal analysis:** Chan Lv.

**Funding acquisition:** Chan Lv.

**Investigation:** Chan Lv.

**Methodology:** Jia Li.

**Project administration:** Chan Lv.

**Resources:** Chan Lv.

**Software:** Jia Li.

**Supervision:** Chan Lv.

**Validation:** Chan Lv.

**Visualization:** Jia Li.

**Writing – original draft:** Jia Li, Chan Lv.

**Writing – review & editing:** Chan Lv.

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
