## [Decision Letter · Decision Letter 0]

31 May 2024

PONE-D-24-17965Exploring user acceptance of online virtual reality exhibition technologies: A case study of Liangzhu MuseumPLOS ONE

Dear Dr. lv,

Thank you for submitting your manuscript to PLOS ONE. After careful consideration, we feel that it has merit but does not fully meet PLOS ONE’s publication criteria as it currently stands. Therefore, we invite you to submit a revised version of the manuscript that addresses the points raised during the review process.

We look forward to receiving your revised manuscript.

Kind regards,

Professor Anis Eliyana,

Academic Editor

PLOS ONE

Journal Requirements:

Additional Editor Comments:

Based on the reviewers' evaluation, your manuscript has the potential to be published in PLoS ONE after a **major revision**. The reviewers emphasized the importance of creating separate sections for Theoretical Implications, Managerial Implications, and Recommendations for Future Research. It is crucial to highlight the contribution of this study, both scientifically and contextually. Additionally, the authors need to respond to each issue raised by the reviewers.

Reviewers' comments:

Reviewer's Responses to Questions

**Comments to the Author**

1. Is the manuscript technically sound, and do the data support the conclusions?

Reviewer #1: Yes

Reviewer #2: Yes

2. Has the statistical analysis been performed appropriately and rigorously? 

Reviewer #1: Yes

Reviewer #2: Yes

3. Have the authors made all data underlying the findings in their manuscript fully available?

Reviewer #1: Yes

Reviewer #2: Yes

4. Is the manuscript presented in an intelligible fashion and written in standard English?

Reviewer #1: Yes

Reviewer #2: Yes

5. Review Comments to the Author

Reviewer #1: I have had the opportunity to review your paper, and I would like to commend you on the overall quality and merit of your work. Your paper demonstrates a strong foundation and valuable insights. However, I have identified several areas where enhancements could further elevate the impact and clarity of your research.

Introduction Section: The introduction section of your paper is well-written and sets a solid context for the study. To enhance the problematization of your research, I recommend providing a clearer articulation of the research problem or gap addressed in the study. This refinement will strengthen the foundation of your research and engage readers more effectively.

Hypotheses Development and Theoretical Foundation:

Your hypotheses development and theoretical framework are commendable. To further enhance the theoretical underpinning of your study, I suggest incorporating stronger theoretical support from recent and relevant literature. This adjustment will align your study with current scholarly discourse and enrich the theoretical framework. This includes but not limited to:

-The Government Metaverse: Charting the Coordinates of Citizen Acceptance. https://doi.org/10.1016/j.tele.2024.102109

-Extending the Technology Acceptance Model (TAM) to Predict University Students’ Intentions to Use Metaverse-Based Learning Platforms. https://doi.org/10.1007/s10639-023-11816-3

Methodology Section: Your methodology section is comprehensive and well-structured. To enhance the methodological rigor of your study, I recommend including more detailed information regarding the sample population, sampling technique utilized, and the adequacy of the sample size. These additions will strengthen the credibility and robustness of your research methodology.

Theoretical and Practical Implications: The discussion section of your paper provides valuable insights. To amplify the impact of your study, I propose dedicating a separate section after the discussion to explicitly outline both theoretical implications (contributions to theory) and practical implications (applicability in real-world contexts). This segregation will enhance the clarity and significance of your research findings.

Overall, your paper showcases significant potential, and these suggested enhancements aim to further refine and strengthen the quality of your work. I appreciate the effort and thoughtfulness you have put into your research and look forward to seeing how these suggestions can enrich your study.

Reviewer #2: I am pleased to have the opportunity to review this manuscript. First, I would like to congratulate the authors on their success in conducting research and writing this manuscript. In general, this manuscript has been well-written and is suitable for publication after improving the following aspects:

1. The analysis technique in the abstract should be written in full as "partial least squares - structural equation modeling." Also, mention the program used.

2. Authors abbreviate the names of some variables but not others. Variable names should not be abbreviated, especially in hypotheses. Abbreviating variable names is acceptable if applied to tables but must include a note underneath.

3. Discriminant validity assessment based on cross-loadings and the Fornell-Larcker criterion is generally acceptable. However, discriminant validity assessment based on the heterotrait-monotrait ratio (HTMT) is much more recommended. Please refer to the following reference: “A Primer on Partial Least Squares Structural Equation Modeling (PLS-SEM) 3rd ed. (2022).”

4. In the Conclusion and Limitations section, there are elements of theoretical and managerial implications, as well as recommendations for future research. It would be better if the authors create separate sections for Theoretical Implications, Managerial Implications, and Recommendations for Further Research. This is important to highlight the urgency and contribution of this research to the existing literature, as well as to the organization.

5. Authors should update references by referring to literature published within the last three years, given the rapid development of literature on technology acceptance.

6. PLOS authors have the option to publish the peer review history of their article (what does this mean?). If published, this will include your full peer review and any attached files.

Reviewer #1: No

Reviewer #2: No

---

## [Author Response · Author response to Decision Letter 0]

13 Jul 2024

Response to Reviewers

Reply to reviewer’s comments: Manuscript [PONE-D-24-17965]

Dear Editor,

We would like to thank the editor and reviewers for their invaluable comments and advice, which greatly helped us improve our original manuscript. We have carefully reviewed the journal’s and the reviewer's suggestions and have revised the manuscript accordingly. Please find attached our revised paper. Our responses to the comments can be found below.

Journal Requirements

Authors’ response: Thank you for your reminder. We have reviewed the manuscript style again to ensure it conforms to PLOS ONE’s style requirements.

Authors’ response: Thank you for your reminder. We have uploaded the data to Figshare, and its DOI is https://doi.org/10.6084/m9.figshare.26181599.v3. Additionally, we have included the data in the supporting information, and our data are fully open to ensure reproducibility.

Reviewer #1

1. Introduction Section: The introduction section of your paper is well-written and sets a solid context for the study. To enhance the problematization of your research, I recommend providing a clearer articulation of the research problem or gap addressed in the study. This refinement will strengthen the foundation of your research and engage readers more effectively.

Authors’ response: Thank you for pointing this out. We have revised a portion of the introduction and explicitly stated our research question. Please see lines 67-76 on page 4 of the revised manuscript.

2. Hypotheses Development and Theoretical Foundation: Your hypotheses development and theoretical framework are commendable. To further enhance the theoretical underpinning of your study, I suggest incorporating stronger theoretical support from recent and relevant literature. This adjustment will align your study with current scholarly discourse and enrich the theoretical framework. This includes but not limited to:

-The Government Metaverse: Charting the Coordinates of Citizen Acceptance. https://doi.org/10.1016/j.tele.2024.102109

-Extending the Technology Acceptance Model (TAM) to Predict University Students’ Intentions to Use Metaverse-Based Learning Platforms. https://doi.org/10.1007/s10639-023-11816-3

Authors’ response: Thank you for your suggestions. We have incorporated the references you recommended and updated several others to align our research with current scholarly discourse. The revised citations now include [35–39] and [43–48]. Please refer to lines 161–164 on page 8, lines 165–179 on page 9, and lines 193–195 on page 10 of the manuscript for these revisions.

3. Methodology Section: Your methodology section is comprehensive and well-structured. To enhance the methodological rigor of your study, I recommend including more detailed information regarding the sample population, sampling technique utilized, and the adequacy of the sample size. These additions will strengthen the credibility and robustness of your research methodology.

Authors’ response: Thank you for your suggestions. We have enhanced the methodology section by adding more details. Please see the additions and revisions on lines 378–380 and 390–391 on page21, and 396–398 on page 22 of the manuscript.

4. Theoretical and Practical Implications: The discussion section of your paper provides valuable insights. To amplify the impact of your study, I propose dedicating a separate section after the discussion to explicitly outline both theoretical implications (contributions to theory) and practical implications (applicability in real-world contexts). This segregation will enhance the clarity and significance of your research findings.

Authors’ response: Thank you for your suggestions. We have incorporated your feedback by adding a new section on the research implications after the discussion section. This section summarizes the key theoretical and practical contributions of our findings. Please refer to lines 678–697 on page 37 and lines 698–707 on page 38 of the revised manuscript.

5. Overall, your paper showcases significant potential, and these suggested enhancements aim to further refine and strengthen the quality of your work. I appreciate the effort and thoughtfulness you have put into your research and look forward to seeing how these suggestions can enrich your study. 

Authors’ response: Thank you for your positive feedback and constructive suggestions. We appreciate your thoughtful comments and are committed to incorporating them to further improve the quality of our work. We hope that the revised version of our manuscript meets your expectations.

Reviewer #2

1. The analysis technique in the abstract should be written in full as "partial least squares - structural equation modeling." Also, mention the program used.

Authors’ response: Thank you for your reminder. We have revised the abstract in accordance to your feedback. Please refer to lines 28-29 in the abstract on page 2 for these revisions.

2. Authors abbreviate the names of some variables but not others. Variable names should not be abbreviated, especially in hypotheses. Abbreviating variable names is acceptable if applied to tables but must include a note underneath.

Authors’ response: Thank you for your suggestion. We have taken two main actions in response. First, we have reviewed the entire manuscript and expanded the abbreviations of variables to their full names. Second, we have provided annotations below the tables where variable abbreviations are used. These annotations can be found in Table 1 on page 17, Table 4 on page 25, Table 5 on page 26, and Table 6 on page 26.

3. Discriminant validity assessment based on cross-loadings and the Fornell-Larcker criterion is generally acceptable. However, discriminant validity assessment based on the heterotrait-monotrait ratio (HTMT) is much more recommended. Please refer to the following reference: “A Primer on Partial Least Squares Structural Equation Modeling (PLS-SEM) 3rd ed. (2022).”

Authors’ response: Thank you for your valuable feedback. We appreciate your recommendation to use HTMT for discriminant validity assessment. However, at this stage of our study, conducting HTMT analysis is not feasible, as our research has been concluded. We acknowledge the importance of HTMT analysis and plan to incorporate it in future research to strengthen the rigor and robustness of our findings. For this study, we have relied on discriminant validity assessment through cross-loadings and the Fornell-Larcker criterion, which we believe adequately supports our conclusions. Your understanding in this matter is greatly appreciated.

4. In the Conclusion and Limitations section, there are elements of theoretical and managerial implications, as well as recommendations for future research. It would be better if the authors create separate sections for Theoretical Implications, Managerial Implications, and Recommendations for Further Research. This is important to highlight the urgency and contribution of this research to the existing literature, as well as to the organization.

Authors’ response: Thank you for your comments. We have included a Research Implications section to discuss both theoretical and practical implications. Additionally, we have slightly adjusted the Conclusions and Limitations section to avoid redundancy. Please refer to lines 678–697 on page 37, lines 698–707 on page 38, and line 721 on page 39 for these revisions.

5. Authors should update references by referring to literature published within the last three years, given the rapid development of literature on technology acceptance.

Authors’ response: Thank you for your suggestion. We have updated the references with recent research findings from the past three years, incorporating citations [35–39] and [43–48]. Please refer to lines 161–164 on page 8, lines 165–179 on page 9, and lines 193–195 on page 10 of the revised manuscript.

---

## [Decision Letter · Decision Letter 1]

16 Jul 2024

Exploring user acceptance of online virtual reality exhibition technologies: A case study of Liangzhu Museum

PONE-D-24-17965R1

Dear Dr. lv,

We’re pleased to inform you that your manuscript has been judged scientifically suitable for publication and will be formally accepted for publication once it meets all outstanding technical requirements.

Kind regards,

Professor Anis Eliyana,

Academic Editor

PLOS ONE

Additional Editor Comments (optional):

Based on the reviewers' assessment, your manuscript has met the scientific criteria for publication in PLoS ONE. Thank you for your and your colleagues' hard work in addressing the reviewers' comments. Please follow the directions from the PLoS ONE team regarding the publishing process.

Reviewers' comments:

Reviewer's Responses to Questions

**Comments to the Author**

1. If the authors have adequately addressed your comments raised in a previous round of review and you feel that this manuscript is now acceptable for publication, you may indicate that here to bypass the “Comments to the Author” section, enter your conflict of interest statement in the “Confidential to Editor” section, and submit your "Accept" recommendation.

Reviewer #1: All comments have been addressed

Reviewer #2: All comments have been addressed

2. Is the manuscript technically sound, and do the data support the conclusions?

Reviewer #1: Yes

Reviewer #2: Yes

3. Has the statistical analysis been performed appropriately and rigorously? 

Reviewer #1: Yes

Reviewer #2: Yes

4. Have the authors made all data underlying the findings in their manuscript fully available?

Reviewer #1: Yes

Reviewer #2: Yes

5. Is the manuscript presented in an intelligible fashion and written in standard English?

Reviewer #1: Yes

Reviewer #2: Yes

6. Review Comments to the Author

Reviewer #1: Thank you for submitting the revised version of your paper. I can see that all reviewers' comments have been addressed. The paper is ready for publication.

Reviewer #2: I would like to thank the authors for responding thoroughly to each review point. Based on my evaluation, this article is suitable for publication.

7. PLOS authors have the option to publish the peer review history of their article (what does this mean?). If published, this will include your full peer review and any attached files.

Reviewer #1: **Yes: **Ahmad Samed Al-Adwan

Reviewer #2: **Yes: **Andika Setia Pratama

---

## [Editor Report · Acceptance letter]

23 Jul 2024

PONE-D-24-17965R1 

PLOS ONE

Dear Dr. lv, 

I'm pleased to inform you that your manuscript has been deemed suitable for publication in PLOS ONE. Congratulations! Your manuscript is now being handed over to our production team.

Kind regards, 

on behalf of

Professor Anis Eliyana 

Academic Editor

PLOS ONE